**Towards an improved representation of carbonaceous aerosols over the**
**Indian monsoon region in a regional climate model RegCM**
Sudipta Ghosh[1], *Sagnik Dey[1,2], Sushant Das[3], Nicole Riemer[4], Graziano Giuliani[3], Dilip
Ganguly[1], Chandra Venkatraman[5], Filippo Giorgi[3], Sachchida Nand Tripathi[6], S.
Ramachandran[7], T.A. Rajesh[7], Harish Gadhavi[7], Atul Kumar Srivastava[8]
[1]Centre for Atmospheric Sciences, Indian Institute of Technology Delhi, India
[2]Centre of Excellence for Research on Clean Air, Indian Institute of Technology Delhi, India
[3]Earth System Physics Section, ICTP, Trieste, Italy
[4]Department of Atmospheric Sciences, University of Illinois at Urbana-Champaign, IL, USA
[5]Department of Chemical Engineering, Indian Institute of Technology Bombay, India
[6]Department of Civil Engineering, Indian Institute of Technology Kanpur, India
[7]Space and Atmospheric Sciences Division, Physical Research Laboratory, Ahmedabad, India
[8]Indian Institute of Tropical Meteorology, New Delhi Branch, India
**\*Correspondence: sagnik@cas.iitd.ac.in**
**Keywords:** RegCM4; emission inventory; carbonaceous aerosols; model customization;
Indian monsoon region
**Abstract.** Mitigation of carbonaceous aerosol emissions is expected to provide climate and
health co-benefits. The accurate representation of carbonaceous aerosols in climate models is
critical for reducing uncertainties in their climate feedback. In this regard, emission fluxes and
aerosol life-cycle processes are the two primary sources of uncertainties. Here we demonstrate
that incorporating a dynamic ageing scheme and emission estimates that are updated for the
local sources improve the representation of carbonaceous aerosols over the Indian monsoon
region in a regional climate model, RegCM, compared to its default configuration. The mean
BC and OC surface concentrations in 2010 are estimated to be 4.25 and 10.35 μg m$^{-3}$,
respectively, over the Indo-Gangetic Plain (IGP), in the augmented model. The BC column
burden over the polluted IGP is found to be 2.47 mg m$^{-2}$, 69.95 % higher than in the default
model configuration and much closer to available observations. The anthropogenic AOD
increases by more than 19 % over the IGP due to the model enhancement, also leading to a
better agreement with observed AOD. The top-of-the-atmosphere, surface, and atmospheric
anthropogenic aerosol shortwave radiative forcing are estimated at -0.3, -9.3, and 9.0 W m$^{-2}$,
respectively, over the IGP and -0.89, -5.33, and 4.44 W m$^{-2}$, respectively, over Peninsular India
(PI). ==Our results suggest that the combined effect of two modifications leads to maximum==
==improvements in the model performance where emissions are playing a dominant role.==

## 1. Introduction

Carbonaceous aerosols (organic carbon, OC, and black carbon, BC) emitted from
incomplete combustion constitute 20%-50% of the total global aerosol mass (Kanakidou et al.,
2005; Putaud et al., 2010), causing substantial air quality degradation (Singh et al., 2021). Due
to their ability to absorb solar radiation, carbonaceous aerosols also contribute to global
warming (Ramanathan and Carmichael, 2008). Hence, they are considered to be key short-
lived climate pollutants (SLCPs) (UNFCC, 2015), and mitigating their emissions is expected
to result in both climate and health co-benefits (Tibrewal and Venkataraman, 2021; Naik et al.,
2021). Climate models are characterized by large discrepancies in simulating carbonaceous
aerosol loadings, their optical properties, and radiative forcing (Ajay et al., 2019), primarily
due to uncertainties in emission inventories and limitations in the treatment of aerosol processes
in the models (Bond et al., 2013). Unless the representation of the life cycle of carbonaceous
aerosols in climate models is improved, their role in climate impacts and air quality degradation
cannot be assessed accurately (Riemer et al., 2019).
A multi-institutional network program - Carbonaceous aerosol emissions, source
apportionment, and climate impacts (COALESCE) was launched by the Government of India
to address some of these issues for the Indian monsoon region (Venkataraman et al., 2020).
One of the scientific objectives of COALESCE is to understand and reduce uncertainties in
representing carbonaceous aerosol life cycle in global and regional climate models, focusing
on the Indian subcontinent. The regional climate model, RegCM4, developed at the
International Centre for Theoretical Physics (ICTP), Italy (Giorgi et al., 2012), is one of the
participating models in COALESCE. RegCM4 was extensively used to examine variability in
the Indian summer monsoon (Dash et al., 2015; Rai et al., 2020), to project climate change over
South Asia (Pattnayak et al., 2018), and to elucidate the dynamical impacts of aerosols on the
Indian summer monsoon in the present (Das et al., 2015, 2016) and future (Das et al., 2020)
climate conditions.
The aerosol module in the RegCM4 (Solmon et al., 2006; Zakey et al., 2006) considers
various aerosol life cycle processes, such as emission (source), advection, horizontal and

vertical diffusion, transport, conversion of hydrophobic to hygroscopic species and wet and dry deposition (sink) (see Methods for more details). Previous studies (Das et al., 2016; Nair et al., 2012) have pointed out that the RegCM4 underestimates the anthropogenic aerosol loading over the Indian subcontinent, and therefore, the net aerosol impact over the region is dominated by natural aerosols (Das et al., 2020). We recently implemented a dynamic ageing scheme in the RegCM aerosol module (Ghosh et al., 2021), which converts carbonaceous aerosols from hydrophobic to hygroscopic states based on the aerosol number concentration. Compared to the constant conversion rate of 27.6 hours used in the default version of the model, the scheme allowed a faster conversion in the polluted regions than in the clean areas of the South Asia region. This, in turn, affected the aerosol forcing due to the changes in aerosol loadings induced by the new hydrophobic-to-hygroscopic conversion scheme. It was also found that implementing the dynamic ageing scheme alone is not sufficient to fully improve the model performance and hypothesized that much of the model uncertainty was due to the emission inventory.

In this work, we examined the changes in carbonaceous aerosol burden and their impact on the radiation budget of the South Asia region due to the combined impact of the improved dynamic ageing scheme and a regional emission inventory (Pandey and Venkataraman, 2014; Sadavarte and Venkataraman, 2014) replacing the global emission inventory used in the default model version (see Methods). We carried out four sets of simulations for the year 2010 - (1) control simulation with the default (fixed) ageing scheme and global inventory (hereafter Default_Sc), (2) simulation with the dynamic ageing scheme and global inventory (Dyn_global), (3) simulation with the default ageing scheme and regional inventory (Fix_Regio) and (4) simulation with the dynamic ageing scheme and regional emission inventory (Dyn_Regio). The changes due to ageing alone (i.e., Default_Sc vs. Dyn_global) have already been reported in Ghosh et al. (2021). Here we analyse and report the improvements in model performance due to the combined impact of incorporating a better emission inventory and a more realistic ageing scheme relative to the default model configuration (i.e., Default_Sc vs. Dyn_regio) and investigate these performance changes in terms of the aerosol processes considered in the model. However, the changes due to emission alone (i.e., Default_Sc vs. Fix_regio) will be considered as an intermediate step towards Dyn_regio.

**2. Data and Methodology**

**2.1 Model configuration**

RegCM version 4 is a hydrostatic, compressible, primitive equation and sigma-p vertical
coordinate model with a dynamical core from the NCAR Mesoscale Model Version 5 (MM5)
(Grell et al., 1994). We have used the Community Climate Model Version 3 (CCM3) (Kiehl et
al., 1996) radiative transfer scheme with the modifications described in the literature (Giorgi
et al., 2012). The model is interactively coupled with both natural (dust and sea salt) (Zakey et
al., 2006, 2008) and anthropogenic aerosols (Solmon et al., 2006), along with a gas-phase
chemistry module (Shalaby et al., 2012), but for this study, we have only considered the
anthropogenic module (Solmon et al., 2006). The choice of parameterisation schemes for our
experiments has been provided in the following table:

| Land surface processes | Biosphere-Atmosphere Transfer Scheme (BATS) (Dickinson et al., 1993) |
|---|---|
| Planetary boundary layer | University of Washington (UW) scheme (Grenier and Bretherton, 2001; Bretherton et al., 2004; O'Brien et al., 2012) |
| Cumulus convection scheme | Emanuel (Emanuel and Živković-Rothman, 1999) over land and Tiedtke (Tiedtke, 1993) over the ocean |
| Large-scale cloud and moisture process | SUBEX scheme (Pal et al., 2007, 2000) |
| Aerosol module | SUCA (Solmon et al., 2006) |
| Emission inventories | IIASA and IIT Bombay 2010 |


The anthropogenic aerosol module consists of sulphate, hydrophilic and hydrophobic BC,
and hydrophilic and hydrophobic OC, along with a sulphate scheme (Qian et al., 2001). The
mass concentrations of these species are tracked, assuming that they form an external mixture.
The emitted carbonaceous aerosols are considered to be 80 % hydrophobic and 20 %
hydrophilic for BC, while equal fractions of hydrophobic and hydrophilic OC are considered
in the simulations. The rate of change of mass mixing ratios of hydrophobic and hydrophilic
tracers, indicated by subscript 'hb' and 'hl,' is described by the chemical transport equation in
Solmon et al. (2006).
The atmospheric lifetime of aerosols is governed by dry and wet deposition. The dry
deposition velocity depends on the type of surface, while the dry deposition flux variation is
proportional to the tracer concentration in the lowest level of the model (around 30 m above
the surface). Wet deposition in the RegCM4 has been split into "in-cloud" and "below-cloud"
terms. The in-cloud removal process starts for large-scale clouds if the liquid water is higher
than the threshold level (0.01 g m$^{-3}$) in the model layers where the cloud fraction is more than
zero and is a function of the fractional removal rate of liquid water (fraction of precipitating
rain over liquid water content of the atmospheric layer, the in-cloud removal rate for cumulus
clouds is constant and fixed at 0.001 s$^{-1}$) and the aerosol solubility. This solubility is different
for different species, and thus hydrophilic and hydrophobic BC/OC have different in-cloud wet
deposition rates. The below-cloud washing out of the aerosols is controlled by their effective
diameters and densities. Collection efficiency for each aerosol species is computed from the
aerosol effective diameter and density, which is different for different species. The changes in
wet and dry deposition alter the ratio of hydrophobic to hydrophilic changes, which in turn
alters the atmospheric lifetime of aerosols. A detailed explanation regarding these changes due
to ageing alone can be referred to Ghosh et al. (2021). Seasonal variation in the lifetime of
particles, at the surface and upper atmosphere, due to ageing alone has been already explained
in Ghosh et al. (2021).
The model was simulated over the South-Asian CORDEX domain (Giorgi et al., 2009) [20°
S - 50° N and 10°-130° E] for the year 2010 at 0.25° × 0.25° resolution, while the results are
analysed over the Indian subcontinent [0°-45° N and 60°-105° E] with special focus on the IGP
[25°-30° N and 73°-89° E] and PI [8°-20° N and 72°-85° E]. The model consists of 18 vertical
levels with 50hPa as the model top pressure. There are three levels (1000, 925, 850 hPa) within
the boundary layer. ERA-Interim reanalysis dataset, at 1.5° resolution and 6-hourly temporal
resolution, has been used to generate the initial and lateral meteorological boundary conditions
for the study (Dee et al., 2011). The sea surface temperature was derived from the NOAA
Optimum Interpolated weekly 1° × 1° gridded data and the chemical boundary conditions from
MOZART 6-hourly data. Four sets of simulations have been performed for the year 2010 - (1)
control simulation with the default (fixed) ageing scheme and global inventory (hereafter
Default_Sc), (2) simulation with the dynamic ageing scheme and global inventory
(Dyn_global), (3) simulation with the default ageing scheme and regional inventory
(Fix_Regio) and (4) simulation with the dynamic ageing scheme and regional emission
inventory (Dyn_Regio). In each of the experiments, the model was simulated from October 01,
2009, to December 31, 2010. The first three months were considered spin-up and thus were not
included in the analysis. The focus of this manuscript is the Indian landmass only. Changes in
aerosol properties over the oceans have not been discussed because the oceanic condition is
mostly clean with low tracer concentration compared to that over the landmass. In the
supplementary material Fig S1, there are hardly any emissions over the oceans. Additionally,
in (Ghosh et al., 2021), it is evident that the ageing time of the carbonaceous aerosols over the
oceans is larger than the default ageing timescale.
**2.2 Emission inventories**
In this study, we considered a global emission inventory
[https://www.iiasa.ac.at/web/home/research/researchPrograms/air/Global_emissions.html]
and a regional emission inventory (Pandey and Venkataraman, 2014; Sadavarte and
Venkataraman, 2014; Venkataraman, 2018). Figure S2 represents the seasonal variation of the
emissions estimated by the two inventories. The global emission inventory used in the
experiments 'Default_Sc' and 'Dyn_global' was developed by the IIASA emission inventory
at a resolution of $0.5° \times 0.5°$
[https://www.iiasa.ac.at/web/home/research/researchPrograms/air/Global_emissions.html].
The key emission sectors considered in this inventory are energy, industry, solvent use,
transport, domestic combustion, agriculture, open burning of agricultural waste, and waste
treatment. The emission estimates were available only at an annual scale with no seasonal
variation from 1990-2010.
The regional emission inventory used in experiments 'Fix_regio' and 'Dyn_regio' was
developed by IIT Bombay (Pandey and Venkataraman, 2014; Sadavarte and Venkataraman,
2014; Venkataraman et al., 2018) at a horizontal resolution of $0.25° \times 0.25°$ and the estimates
vary at a monthly scale. Thus, the regional emissions have a profound seasonal variability
(Figure S1). The key sectors included in the regional inventory are energy (coal + oil + gas),
heavy and light industry, brick production, residential cooking, solid biomass fuel, residential
cooking (LPG and kerosene), residential lighting (kerosene lamp), residential water heating,
residential space heating, informal industry, agricultural residue burning, on-road gasoline, on-
road diesel, railway, agricultural diesel pump, agricultural tractors. Among these sectors,
residential water heating, residential space heating, and agricultural residue burning sectors
have seasonality in emissions.
**2.3 In-situ BC data**
In-situ BC data for the year 2010 has been procured from 24 sites to evaluate the model
performance. These sites have been shown in the supplementary Fig S4. 21 of these sites are
part of the Indian Space Research Organization's Aerosol Radiative Forcing over India
Network, ARFINET (Babu et al., 2013; Gogoi et al., 2021). This network has been measuring
columnar AOD and BC for many years. In addition to the ARFINET, BC concentrations are
also measured independently at Kanpur (Tripathi et al., 2005) (entire 2010 except during the
monsoon season), Gadanki (Gadhavi et al., 2015; Jain et al., 2018), and Delhi (October-
December 2010) by individual institutions. In all the sites, BC was measured by an
Aethalometer. An aethalometer measures the amount of attenuation of the light beam passing
through the filter where particles get deposited. BC mass concentration is measured by the
change in optical attenuation given by the rate of BC deposition on the filter tape (Hansen et
al., 1984). Dataset from all the sites except Gadanki (monthly values) are available on a daily
scale and have been averaged to get the annual concentrations.

**2.4 MERRA-2 data**
Model simulated BC and OC columnar burdens have been evaluated against MERRA-2
reanalysis data. MERRA-2 is an updated reanalysis of atmospheric data produced by the NASA
Global Modeling and Assimilation Office (Buchard et al., 2017). MERRA-2 consists of
parameters that are not available in its predecessor, MERRA. It includes updates of the
Goddard Earth Observing System model and analysis scheme in order to give a more realistic
view of the ongoing climate analysis beyond MERRA's jargon. This dataset addressed the
limitations of MERRA. Various improvements in MERRA-2 include assimilation of aerosol
observations and improved representation of stratosphere, including ozone and cryosphere.
MERRA-2 data products are freely accessible through the NASA Goddard Earth Sciences Data
Information Services Center. We note that MERRA-2 data are also not observations and direct
validation of the MERRA-2 columnar BC and OC burden is not possible.

**2.5 MISR aerosol data**
MISR on-board Terra satellite crosses the equator around 10:30 hrs local time. It has a high
spatial resolution and a wide range of viewing angles. It views the Earth using four spectral
bands in each of the nine cameras and has a weekly global coverage between ±82º. A detailed
description is provided in the literature (Diner et al., 1998). MISR-AOD has a correlation
coefficient of ~0.9 (for maritime sites) and ~0.7 (for dusty sites) w.r.t AERONET (Kahn et al.,
2005). In the absence of any direct measurement of anthropogenic AOD, we use MISR fine
AOD (AOD for particles smaller than 0.35 $\mu$m) (Dey and Di Girolamo, 2010).

## 3. Results

In this section, we have discussed the three-dimensional annual distribution of carbonaceous aerosols (sections 3.1 and 3.2) for the default (Default_Sc) and augmented (Dyn_regio) model set-up. The seasonal distributions for all four experiments – Default_Sc, Dyn_global, Fix_regio, and Dyn_Regio, have been reported in the supplementary information (SI). In section 3.3, we have investigated the annual changes in aerosol optical properties due to the default (Default_Sc) and augmented (Dyn_regio) model set-up. In this case, also, the seasonal variability across the four experiments has been shown in the SI.

### 3.1 Spatial distribution of carbonaceous aerosols

Figure 1 shows the spatial distributions of the annual surface concentration for BC and OC using the default and augmented model, along with their differences. Several key features are notable. First, the OC concentration is almost three times higher than the BC concentration in the augmented model, consistent with the literature (Priyadharshini et al., 2019). Secondly, the concentrations are 2-3 times higher over the polluted Indo-Gangetic Plain (IGP) compared to the rest of India in the augmented model. High aerosol loadings in the IGP are a result of the combined effects of greater source strength, low topography surrounded by highlands to the north and south, and unfavourable meteorology (Dey and Di Girolamo, 2010; Srivastava et al., 2012). Thirdly, the BC and OC concentrations increase by >100 % and >60 %, respectively, over the IGP and by smaller margins elsewhere in the augmented model relative to the default configuration. The increase in the annual tracer concentrations can be further explained by the seasonal distributions and the selected model configuration. To begin with, an increase in both BC and OC concentrations during the winter (JF), pre-monsoon (MAM), and post-monsoon (OND) seasons are clearly visible in Fig S2 and Fig S3 (see Supplementary Information SI). During the monsoon, precipitation removes large amounts of aerosols; as a result, the increase in BC concentration is almost negligible, and for OC, it is negative. The transition in concentration from Dyn_global to Fix_regio is most prominent than that from Default_Sc to Dyn_global or Fix_regio to Dyn_regio. This indicates the impact of the switch from the global to regional emission inventory (Figure S1) is greater than the impact of the implementation of the dynamic ageing scheme (Figure S1) on the increases in BC and OC mass concentrations in the augmented model.

253

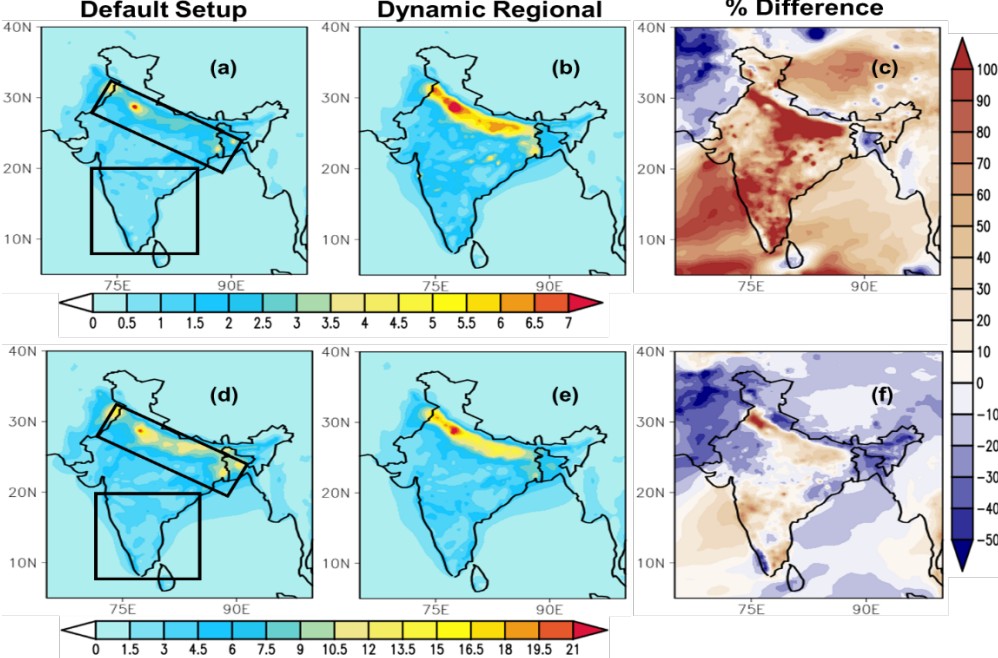

**Figure 1** Spatial distribution of surface mass concentration (µg m⁻³) of BC (a, b) and OC (d, e) in 2010 over the Indian subcontinent using (left) the default and (middle) the augmented model configurations. Figures 1c and 1f represent the corresponding percentage differences due to the augmented model set-up (positive values imply an increase in mass concentration). The vertical distributions (shown in Figure 3) are analysed for the IGP and PI sub-regions marked by boxes in the panels of the left column.

We evaluate the performance of the customized model against BC surface concentrations measured at 24 sites across India (Figure 2). We note that the in-situ concentrations are point measurements, while the model grids containing these sites are representative of 25 km by 25 km areas. The default model severely underestimates the surface BC compared to the in-situ observations (mean normalized bias, MNB = -69 %). Though the underestimation persists in the augmented model (by varying proportions across the sites), the simulated concentration magnitudes are closer to the observations (MNB = -51 %), particularly in the mega-cities of the polluted IGP (e.g., Delhi, Kanpur, Varanasi, Kharagpur). The improvement is small in some cities, particularly in the East India region (e.g., Dehradun, Dibrugarh, Ahmedabad), where the differences in global and regional emission inventories are also small. This suggests that the problem could be related to the emission fluxes. In several cities, especially in the North and

South Indian regions (e.g., Goa, Nainital, Ooty, Thiruvananthapuram), the simulated BC using
the augmented model is a very close match with the observations. Overall, the augmented
model ($R^2 = 0.66$) performs better than the default model ($R^2 = 0.6$) in simulating surface BC
concentrations, and the errors shown in Figure 2 could also be amplified by the fact that the
model data refers to a 25 km $\times$ 25 km area as a single grid.

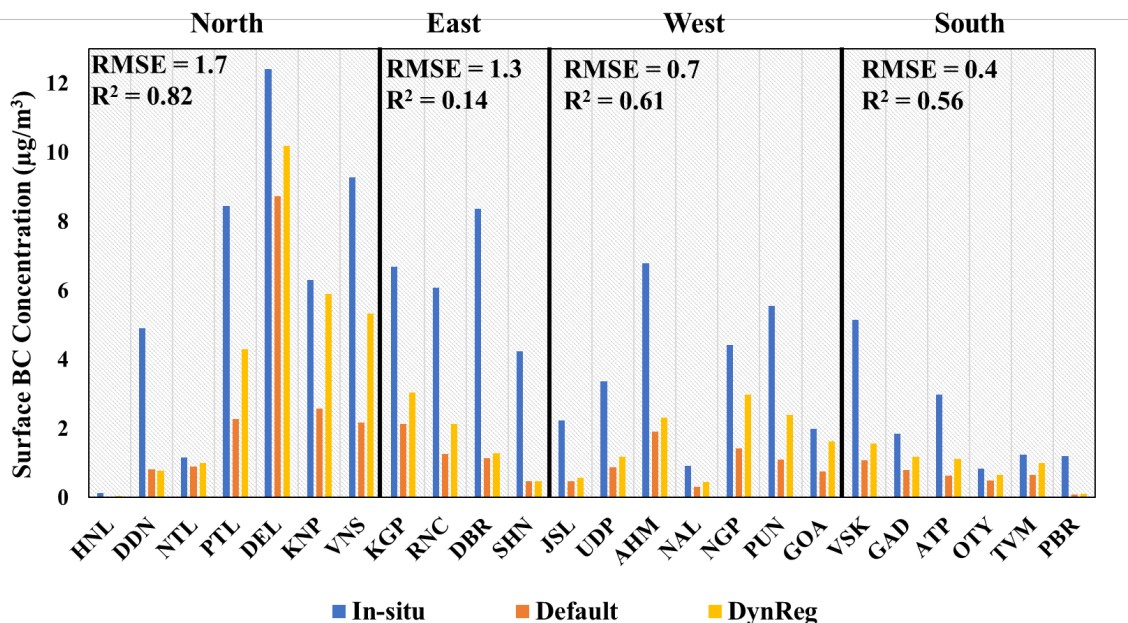



**Figure 2** Comparison of simulated BC surface concentration ($\mu$g m$^{-3}$) using the default and
augmented model with in-situ measurements from 24 cities across India. Locations of the cities
are shown in Figure S4. RMSE (in $\mu$g m$^{-3}$) and $R^2$ between the customized model simulations
and surface measurements are also provided.

287       Since there are no in-situ measurements of columnar burden, we compare the simulated

columnar burden (Figure 3) with data from Modern-Era Retrospective Analysis for Research
and Applications, version 2 (MERRA-2) (Buchard et al., 2017). Similar to the surface mass
concentration, the BC burden shows a more pronounced change than the OC burden due to the
inclusion of the new model features. Here also, the introduction of the emissions alone played
a more prominent role than the dynamic ageing alone (see Figure S5 and Figure S6 in SI), but
the highest change can be observed in the presence of both. Though the simulated burden is
still underestimated relative to the MERRA-2 data, the values in the augmented model are
much closer to the reanalysis data, and the sequence of changes (in both BC and OC) follow
Dyn_regio > Fix_regio > Dyn_global > Default_sc. During the winter season (Jan-Feb), the
percentage difference of model-simulated column burden (w.r.t. MERRA-2) decreases from
>70 % to ~ 35 % for BC and from ~63 % to ~49 % for OC in the augmented model (Figure S5
and S6). A similar improvement is found in the pre-monsoon season (Mar-May). The higher
BC loading over the IGP results from higher magnitudes of regional emissions coupled with
faster ageing and slower removal rate. The percentage difference increases for OC burden over
northwest India decreases over the IGP and is negligible over the rest of the country. A probable
explanation for such OC distribution relies on the emission inventories used since the OC
emissions are slightly higher in the global inventory than those in the regional inventory over
northwest India and lower in the IGP. Emissions over the PI are roughly similar in the two
inventories (Figure S1). The dominant role of emissions in both BC and OC simulated burden
is further supported by the observed transition changes from Dyn_global to Fix_regio. We also
note that anthropogenic aerosol emissions vary on an annual basis in MERRA-2 (Buchard et
al., 2017); hence, there could be larger uncertainties at a seasonal scale.
During the monsoon season (Jun-Sep), the BC loading increases, and OC loading decreases
in magnitude in the augmented model compared to the default set-up (Figure S5), mostly due
to the implementation of the regional inventory. The magnitude of the simulated BC column
burden is comparable between the Default_Sc and Dyn_sc experiments and that between the
Fix_Regio and Dyn_Regio (Figure S5), with an opposite pattern found for the OC column
burden (Figure S6). Two possible reasons can explain this result. First, the OC emissions in the
global inventory are higher than in the regional ones (Figure S1). Second, the model assumes
that OC is 50 % hydrophobic and 50 % hydrophilic at the time of emission (for BC, it is 80 %
hydrophobic and 20 % hydrophilic), and therefore the faster conversion to hydrophilic OC due
to the dynamic ageing can enhance the hydrophilic OC removal by rain. On analysing the wet
removal (refer to Figure S7 and Figure S8 in SI), BC_HL showed the expected highest removal
during JJAS, but OC_HL showed a lower magnitude of wet removal. Therefore, lower OC
emissions in the regional inventory play a major role, during JJAS, in simulating OC burden in
the augmented model. In the post-monsoon season (Oct-Dec), an overall increase in column
burden in the augmented model is observed throughout India. Higher emissions (in the case of
the regional inventory) result in higher concentrations of available condensing and coagulating
particles, which in turn allows faster ageing of hydrophobic to hydrophilic BC leading to
accumulation of BC particles in the atmosphere before their removal by dry deposition. The
changes in the OC loading are negligible in this season (refer to Figure S6 in SI).

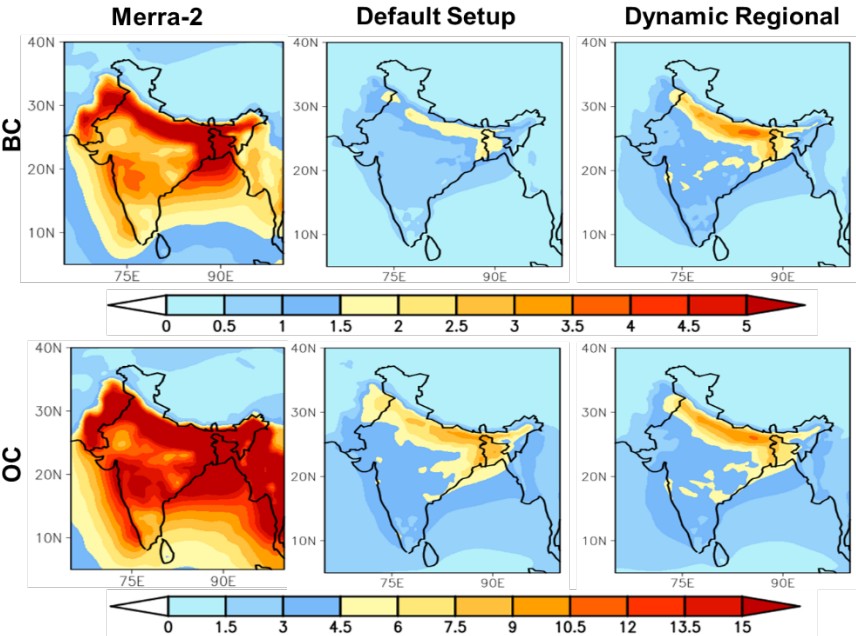


**Figure 3** Comparison of spatial patterns of annual (top panel) BC and (bottom panel) OC
column burden (mg m$^{-2}$).

## 3.2 Vertical distribution of carbonaceous aerosols

In this section, we analyse the effects of the model improvements on the vertical
distribution of aerosols over the IGP and compare the results with the contrasting PI region,
where the emissions are much lower. The two regions are indicated by the boxes in Figure 1.
Figure 4 represents longitude-altitude cross-sections of annual BC and OC mass concentration
(μg m$^{-3}$) over the regions. The vertically distributed mass concentrations (μg m$^{-3}$) of both BC
and OC increase due to the model improvements up to 500 hPa. Similar to spatial distribution,
here also seasonal variability will help to explain the annual vertical concentrations.
Furthermore, the changes are more dramatic and prominent over IGP than that over PI.

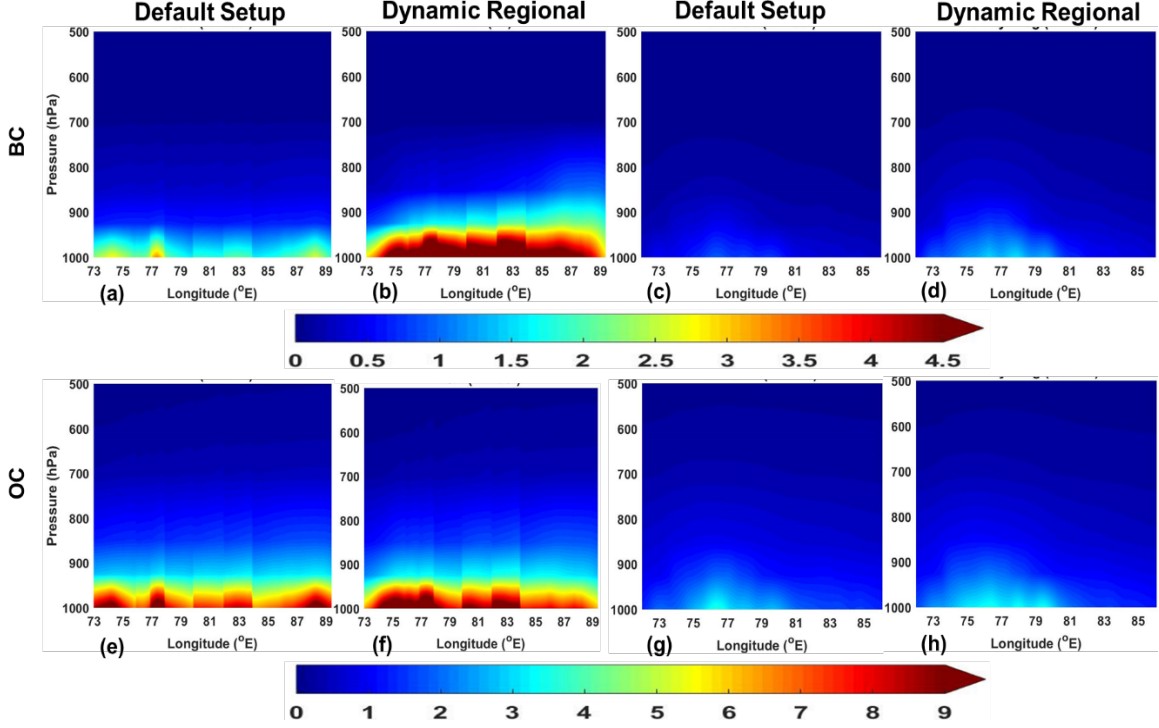

**Figure 4** Longitude (in °E)-altitude (in hPa) cross-sections of (top panel) BC and (bottom panel) OC mass concentration ($\mu g\ m^{-3}$) over the IGP (a, b, e, f) and PI (c, d, g, h) for the default and customized model.

Over IGP, a larger increase is observed during the winter and post-monsoon seasons (Figures S9 and S10). Both the BC and OC concentrations ($\mu g\ m^{-3}$) are comparable in the default and Dyn_global configurations, but they increase in the Fix_regio and Dyn_regio set-ups (Figures S9 and S10). This observation reflects the pre-dominant role of regional emissions. In these two seasons, both BC and the OC are distributed up to the mid-tropospheric levels but with differing magnitudes. This is indicative of higher concentrations of OC vertical transport than that of BC. During the pre-monsoon season, the vertical distributions of both BC and OC show responses similar to that of their spatial distributions. In the monsoon season, the tracer concentration is mainly confined to the surface levels, indicating a lower wet removal and slower ageing above 1000 hPa (Ghosh et al., 2021).

For further clarification of the seasonality of the vertical pumping effect, the convective tendency (represents vertical transport) and lateral advection (represents horizontal transport) have been investigated. The model simulated convective tendency and lateral advection (responsible for long-range transport) are given below. More positive values indicate a strong updraft above the surface due to convection. Convection tendency gradually increases from

left to right (Figure 5 for BC and Figure S11 in SI for OC). Particularly in the drier seasons
since more particles are available in the absence of washout. During winter, the augmented
model (Dyn_regio) shows a lesser pumping effect over IGP than that when only emissions
have been changed (Fix_regio). This can be due to the fact that in the presence of dynamic
ageing a greater number of hydrophilic tracers are available for removal (evident from the
removal plot of BC_HL) even for a small amount of precipitation from western disturbances.
However, during post-monsoon (OND), due to negligible precipitation over IGP, removal rates
of hydrophilic tracers are comparable, and hence the pumping effect also follows the same
trend. A similar trend in convective tendency is also shown by OC particles (Figure S11 in SI).
The magnitude of OC convection tendency is stronger than that of BC particles, probably due
to a higher concentration (Priyadharshini et al., 2019; Ram et al., 2010) of available particles.
Besides, lateral advection is an indicator of horizontal long-range aerosol transport. More
positive values indicate strong flow along the surface due to advection. Advection shows strong
seasonality (from top to bottom – Figure 6 for BC and Figure S12 in SI for OC). In drier months
(JF and OND), horizontal transport is comparatively less than in pre-monsoon (MAM) and
monsoon (JJAS). Therefore, vertical convection is more prominent in dry seasons while
horizontal advection is dominant for MAM and JJAS, irrespective of the choice of schemes.
Consequently, the observed BC concentration is due to convection in JF and OND and due to
advection in MAM and JJAS. Similar logic can be applied for OC concentration distribution
due to lateral advection (Figure S12 in SI). However, the positive advection signal is stronger
than that of BC particles. This can be again due to the higher concentration of available particles
for transport to other regions.

386       In addition, the atmospheric profiles over the region have also been used to explain the tracer

distribution. In terms of changes in temperature profile, higher temperatures over IGP during
MAM and JJAS facilitated the strong vertical wind movement (negative values in Figure S13).
But negative convective tendency (Figure 5 for BC and Figure S11 in SI for OC) and positive
lateral advection of (Figure 6 for BC and Figure S12 in SI for OC) carbonaceous aerosols
during these months lowered their concentrations. This is further supported by the high RH
values particularly in JJAS (Figure S14) which resulted in higher removal. Exactly opposite is
happening during the drier months (JF and OND). Comparatively low temperatures (Figure
S15), facilitated more stable wind movement (positive values in Figure S13). However, in
presence of high emissions, the aerosol pumping effect resulted in strong convective tendency
(Figure 5) which further facilitated the higher concentrations during these months. The low RH

values (Figure S14) during these months are also conducive of higher aerosol atmospheric lifetime.

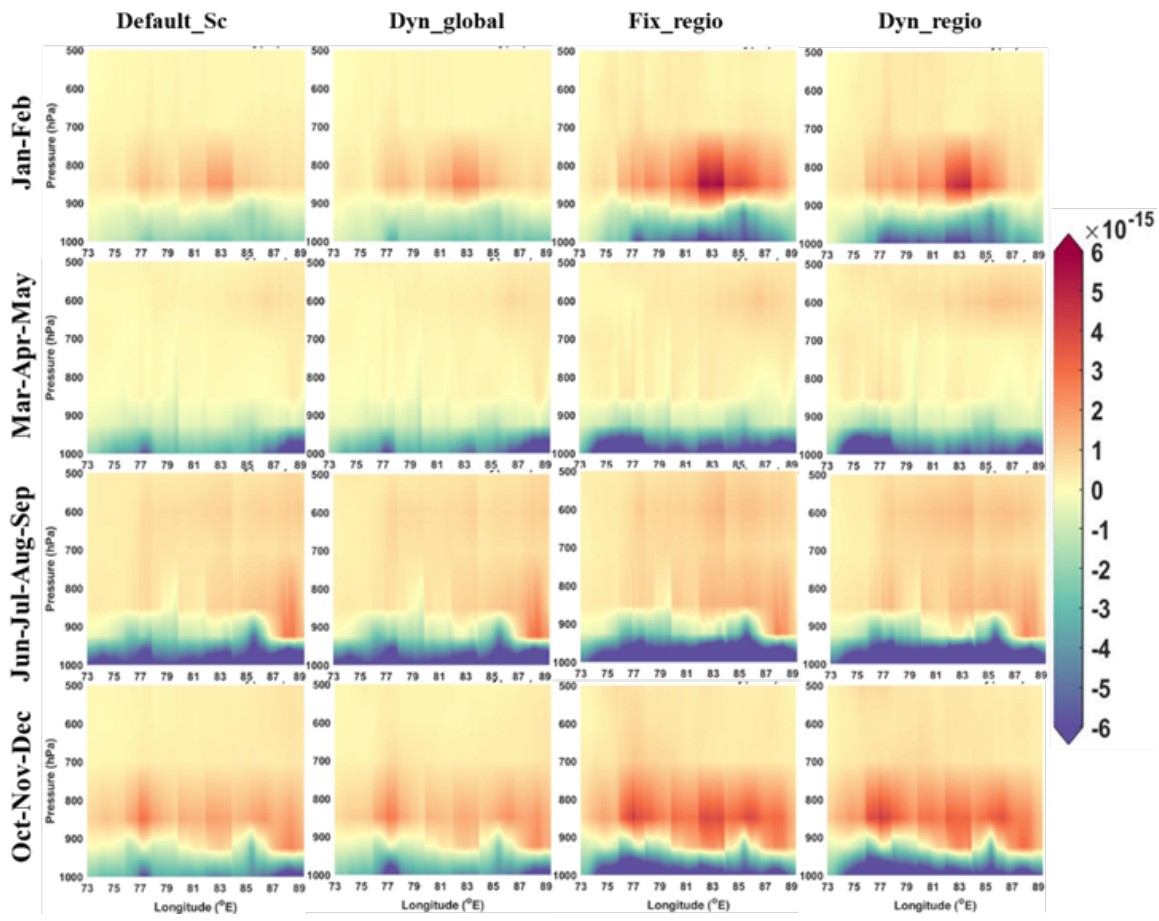

**Figure 5** Seasonal distribution of convective tendency (kg kg$^{-1}$ s$^{-1}$) of BC over IGP for four distinct experiments.

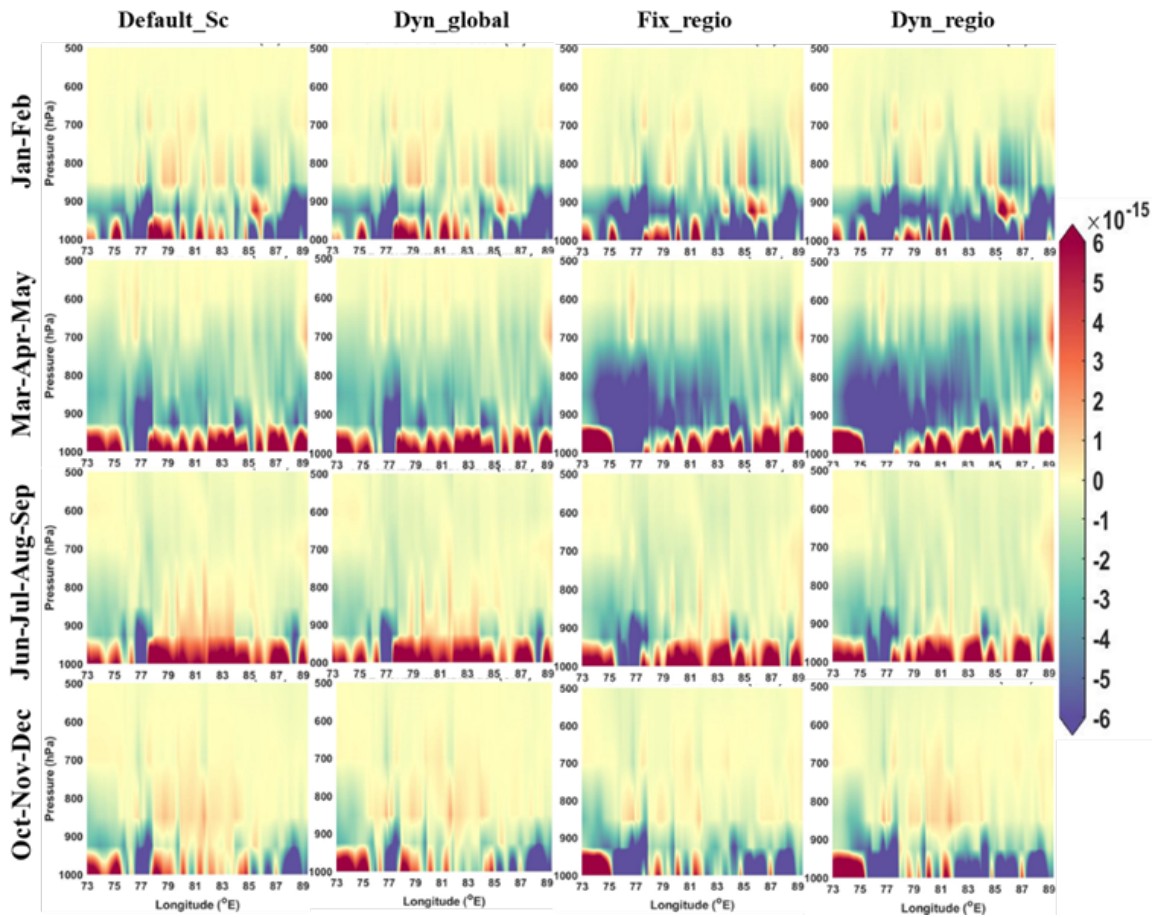

**Figure 6** Seasonal distribution of lateral advection (kg kg$^{-1}$ s$^{-1}$) of BC over IGP for four distinct experiments.

Over the PI, the annual concentrations of carbonaceous aerosols (Figure 4) are very low than over the IGP, which limits the impact of dynamic ageing because of the lower availability of condensing and coagulating particles (relative to the IGP). This results in a slower ageing and lower accumulation of hydrophilic tracers in the troposphere. However, the vertical pumping effect is quite prominent during the winter season in the augmented configuration (Figures S16 and S17). During the pre-monsoon season, only the BC concentration shows an increment in the lower troposphere, while the OC concentration remains more or less unchanged. The PI receives rainfall during the southwest and northeast monsoon; hence the tracer concentration is further lowered during the monsoon and post-monsoon seasons. This is further supported by the high relative humidity values over PI during monsoon and post-monsoon (Figure S18). The high humidity during JJAS can also influence a comparatively high, near surface air temperature (Figure S19) by trapping the radiation. This in turn resulted

in a high vertical wind shear (Figure S20) over PI during this season. But the convective
tendency is low for both BC and OC (Figure S21 and S24 respectively).
The lower concentration can be, therefore, primarily because of the lower emissions for both
BC and OC (refer to Figure S1). This argument is further supported by lower washout than IGP
(Figure S7 and S8) in spite of high RH values (Figure S18). Since, the convective tendency, as
well as lateral advection for BC, is not playing any major role (as can be seen in Figure S21
and Figure S22 in SI), therefore again concluding the role of lower emissions. In the case of
OC, lateral advection (Figure S23 in SI) and comparatively lower emissions (Figure S1 in SI)
than IGP can be the predominant factors for lower concentration over PI in the presence of
negative convective tendency (Figure S24 in SI).

**3.3 Optical and radiative properties of anthropogenic aerosols**
We now examine the effects of the model improvements on the optical properties of
anthropogenic aerosols. In this regard, we note that the changes due to the implementation of
the dynamic ageing scheme can alter only BC and OC concentrations, while the changes related
to the emission inventory impact the sulphate concentration as well. We consider the AOD due
to small particles (radius<0.35 μm) from the Multiangle Imaging Spectroradiometer, MISR
(Kahn and Gaitley, 2015), as a proxy for anthropogenic AOD (hereafter AAOD) since direct
measurement of AAOD are not available to evaluate our model performance (Figure 7).
The simulated annual AAOD is >50 % lower than the MISR small-AOD over the polluted
IGP and 30-50 % lower over the PI in the default model. This is consistent with the previous
studies (Nair et al., 2012). These model underestimations improve by 25-35 % over the IGP
and parts of PI in the augmented model. The seasonal plots (Figure S8) clearly show an increase
in AAOD in all seasons except during the monsoon. This increase in AAOD is due to both the
implementation of region-specific emission fluxes (Nair et al., 2012) and the dynamic ageing
scheme (Ghosh et al., 2021). The AAOD still remains underestimated in some regions, which
can possibly be addressed by further improvements of the emission estimates, for example, the
addition of missing sectors (e.g., crematorium, municipal solid waste burning, etc.), improving
sectoral methodologies for informal activities and incorporation of regionally measured
emission factors.

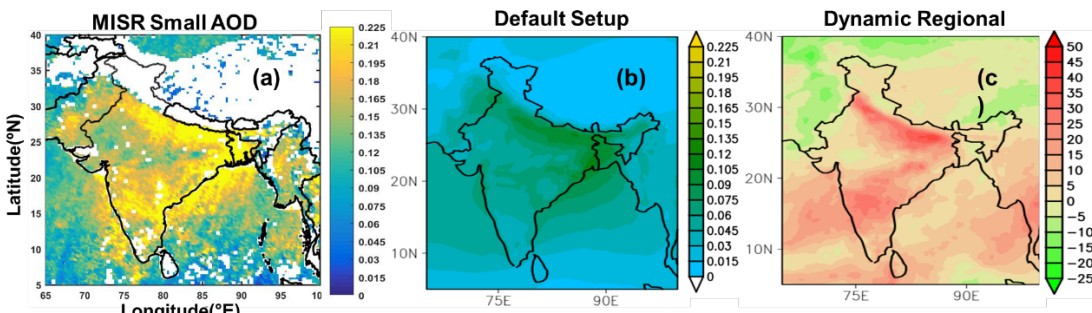


**Figure 7** Spatial distribution of (a) MISR small mode AOD ('white' color implies 'no data'), (b) AAOD simulated by default_sc, and (c) percentage increase in AAOD simulated by the augmented model w.r.t default_sc for 2010.

Spatial patterns of the annual top-of-the-atmosphere (TOA), surface, and atmospheric radiative forcing associated with the anthropogenic aerosols for the augmented model are shown in Figure 8. Currently, the model does not assume aerosol interaction with clouds; therefore, the radiative feedback is mainly governed by direct radiative forcing. Hence, secondary effects due to aerosols cannot be considered for the observed values in Fig 8. The TOA aerosol radiative forcing lies in the range of -0.5 to -1.5 W m$^{-2}$ over most of the Indian landmass, except the IGP, where it is positive (0.25 to 1 W m$^{-2}$) due to the higher concentration of carbonaceous aerosols (Figure 1 and Figure 3), particularly BC. The TOA forcing is also positive over the Indian desert and snow-covered regions even when the carbonaceous aerosol concentrations are lower or comparable to the rest of India. The high surface albedo in these regions allows for an enhanced interaction of the carbonaceous aerosols with solar radiation, resulting in a warming effect (Satheesh, 2002). The surface radiative forcing is found to be larger than -10 W m$^{-2}$ over the polluted IGP, which is consistent with published results (Ramanathan and Carmichael, 2008). Over the rest of India, the surface forcing values lie between -3 to -8 W m$^{-2}$. Due to the model improvements (forcing estimates with the default model are shown in Figure 8), the TOA forcing changes by -72.75 %, and the surface dimming increases by 39.73 % over the IGP and by -23.94 % and 34.35 %, respectively, over PI. As a result, the atmospheric heating increases by ~9 W m$^{-2}$ over the IGP. The simulated surface shortwave radiation shows a statistically significant (p<0.05) correlation with the observations from CERES (Su et al., 2005) all-sky and clear-sky radiation throughout the year except in MAM and JJAS clear-sky conditions (Figure S26 and S27). Here we didn't separate clear and cloudy days because the aerosol-cloud interactions are absent in the model. Therefore, the

reflection from clouds will also be lower. As a result, contribution to the observed AAOD (in
supplementary figure S25) due to cloud reflections will also be lower. Therefore, AAOD
distribution over IGB is primarily responsible for the surface dimming effect and the resulting
atmospheric heating.

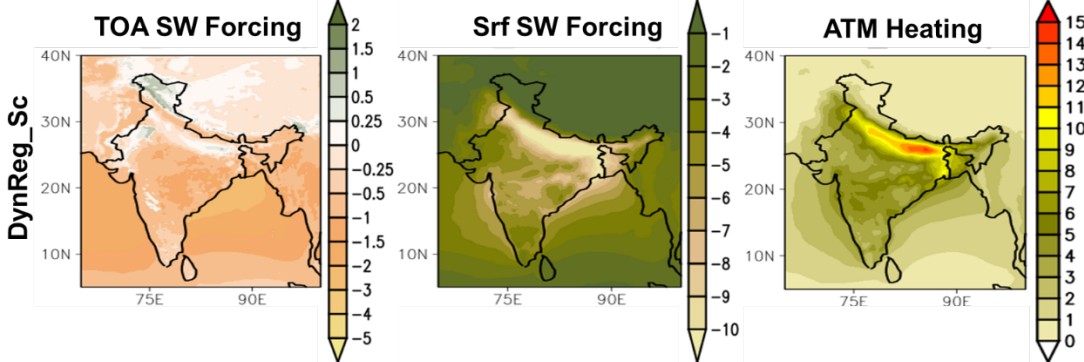


**Figure 8** Annual variation of SW radiative forcing (W m$^{-2}$) at TOA (left column), at the surface
(middle column), and the resultant atmospheric heating (W/m$^2$) (right column) for the
customized set-up.


## 4 Discussion and conclusions

Accurate estimates of emission fluxes and a better representation of aerosol processes are
required to improve the representation of aerosol life-cycle and radiative effects in climate
models. Here we modified the regional climate model RegCM4 by implementing a dynamic
ageing scheme and a regional emission inventory and examined the combined impact of these
factors on the model performance over the Indian monsoon region. Percentage difference in
the figure 9, clearly showed that experiment Fix_regio is simulating comparable BC
concentration and burden (except monsoon) as that by Dyn_regio (augmented model).
Therefore, regional emission is acting as a dominant influencer in the model estimates of tracer
distribution. We note that though the aerosol simulations improve due to these model
enhancements, some systematic biases persist (underestimation of carbonaceous aerosol
concentrations) and need to be further addressed. For example, RegCM has a bulk scheme for
anthropogenic aerosols, and thus the number concentration is calculated from the bulk mass
concentration (Ghosh et al., 2021). The anthropogenic aerosol module can thus be improved
by including a particle size-dependent representation. In addition, the present dynamic ageing
timescale depends only on the anthropogenic aerosol number concentration, while it should, in

fact, depend on the total (anthropogenic + natural) number concentrations. The simulations presented in this work did not include natural aerosols, which could have impacted the meteorology through dynamic feedback, possibly affecting the carbonaceous aerosol burden. This aspect will be examined in future work. Thirdly, though the emission fluxes of BC, OC, and $SO_2$ are higher in the region than the global inventory, there may still be uncertainty related to missing sectoral sources.

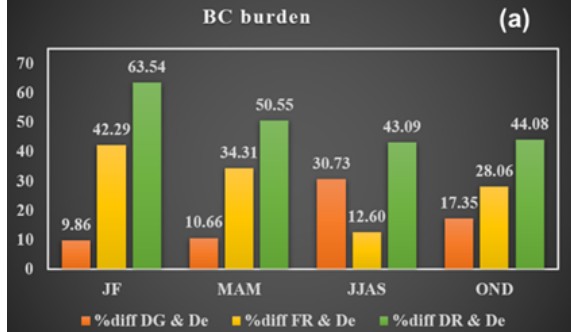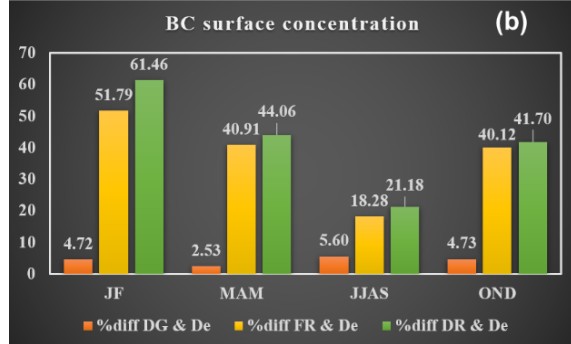

**Figure 9:** Seasonal variations of percentage changes of (a) BC burden (mg/m$^2$) and (b) BC surface concentration (µg/m$^3$) for each sensitivity experiment w.r.t the default set-up where De = Default, DG = Dyn_global, FR = Fix_regio and DR = Dyn_regio.

Our work demonstrates that even the improvement of some aspects of the aerosol representation can lead to substantial enhancements in the model performance. We also find that over the South Asian monsoon region, particularly over highly polluted regions such as the IGP, the default model significantly underestimates the surface dimming and atmospheric heating, which can have implications for climate studies (Das et al., 2016, 2020) and this problem is substantially ameliorated with our model augmentations.

The key conclusions of our work can be summarized as follows.

1. The conclusion in the model RegCM4 implementation of a dynamic ageing scheme and a regional emission inventory substantially improves the model performance over the Indian sub-continent.

2. Combined impact of both modifications leads to improvements on the model performance, in simulating BC and OC surface concentration and column burden. However, the emissions are playing a dominant role.

3. The TOA, surface, and atmospheric radiative forcing are estimated to be -0.3, -5.3, and 5.0 W m$^{-2}$, respectively, over the polluted IGP using the augmented model, but they could still be underestimated.

*Data availability.* The model RegCM4 code is freely available online from (https://gforge.ictp.it/gf/project/regcm/). The anthropogenic aerosol emissions considered for the simulations are taken from the IIASA inventory. The data used can be easily accessed online at http://clima-dods.ictp.it/Data/RegCM_Data/RCP_EMGLOB_PROCESSED/iiasa/ website. Input files for the RegCM4 model are archived on http://clima-dods.ictp.it/Data/RegCM_Data/ website. MISR data is available freely from https://www-misr.jpl.nasa.gov/ while MERRA-2 data is freely available from the NASA Giovanni site https://giovanni.gsfc.nasa.gov/giovanni/.

*Competing Interests.* All the authors declare that they have no conflict of interest.

*Acknowledgements.* We thank the Aerosol Radiative Forcing over India (ARFINET) project of ISRO GBP for sharing the BC data. The authors thank the internal review committee of the NCAP-COALESCE project for their comments and suggestions. The views expressed in this document are solely those of the authors and do not necessarily reflect those of the Ministry. The Ministry does not endorse any products or commercial services mentioned in this publication. SG acknowledges the supercomputing facility Keeling of the University of Illinois Urbana-Champaign. SD acknowledges IIT Delhi for the support for the Institute Chair fellowship.

*Financial Support.* This work is supported by the MoEFCC under the NCAP-COALESCE project [Grant 14/10/2014-CC]. SG acknowledges the support for the DST-INSPIRE fellowship (IF150055) and Fulbright-Kalam Climate Doctoral Fellowship. NR acknowledges funding from NSF AGS-1254428 and DOE grant DE-SC0019192. Funding from the Department of Science and Technology – Funds for Improvement of Science and Technology infrastructure in universities and higher educational institutions (DST-FIST) grant (SR/FST/ESII-016/2014) is acknowledged for the computing support.

564

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
