# Peer review of "Towards an improved representation of carbonaceous aerosols over the"

_Geoscientific Model Development, 2021_

## Referee Comment (RC1)

Towards an improved representation of carbonaceous aerosols over the Indian monsoon region in a region climate model RegCM4.6

Ghosh et al.,

General comments:
 A set of sensitivity studies from a changes to emissions and the ageing factor (conversion rate of hydrophobic to hydrophilic) were presented using RegCM4.6 for BC and organic carbon aerosols over India.  This was shown to produce better results and reduce the bias often by factors of two or more over much of India.  Most of the discussion and improvements in model performance are presented in terms of default simulations and the soc-called dynamic regional, that includes adjustments to both the emissions and ageing.  It is thus hard to understand if most of the improvements are due to emissions or due to changes in the ageing. As the authors are aware there are numerous papers on sensitivity of the aerosol burden over India to emissions and that is not worth another paper as it adds no new information beyond what we already know. The ageing changes should change the ratio of dry deposition/wet deposition in the model and later the lifetime of the aerosols in the model.  There are no figures in the paper that show the deposition fluxes and their changes and I am left to wonder if that was even significant in the results? As the primary idea seems to be that this will change the lifetime of BC and OC in the model, why are there no calculations of lifetime of particles regionally, seasonally and annually?
Are there any changes because of the lifetime changes in particle fluxes over to Indian Ocean that varies by season?
The model setup also leaves much to be desired. There are only 17 levels in the models, not sure how many of them are in the boundary layer? How well does the model simulate convection and mixing over the Ganges valley and central plains where the transport to above PBL could be a major factor in increasing the long-range transport of aerosols and hence their lifetimes? How well does the model simulate column water depths and hence removal rates through wet deposition in the model?
The radiative forcing calculations are for direct radiative forcing one assumes as there is no discussion of aerosol-cloud interactions in the model. If that is the case, did you separate the clear days from cloudy days to perform these forcing calculations are these are seasonal averages for days with and without clouds?  How well does the model represent RH in the vertical column (Feng et al.,RAWEX-GVAX special issue, Current Science, 2016). How much effect does the change in aerosol burden in the column have on the atmospheric profiles and how much of this contributes to changes in optical properties and hence forcing?
Overall, I find the manuscript poorly constructed and doesn't add any new insights to what is already known and needs a rethinking from the authors on both the analysis and experimental design.

I should also point out that the figures are poorly constructed with figure titles that are not very descriptive, figure 2 for instance the title has misspelled 'South' as 'Sout'.

---

## Author Response (AR1)

**"A set of sensitivity studies from a changes to emissions and the ageing factor (conversion rate of hydrophobic to hydrophilic) were presented using RegCM4.6 for BC and organic carbon aerosols over India. This was shown to produce better results and reduce the bias often by factors of two or more over much of India. Most of the discussion and improvements in model performance are presented in terms of default simulations and the soc-called dynamic regional, that includes adjustments to both the emissions and ageing. It is thus hard to understand if most if the improvements are due to emissions or due to changes in the ageing the authors are aware there are numerous papers on sensitivity of the aerosol burden over India to emissions and that is not worth another paper as it adds no new information beyond what we already know."**

> _**Response:**_ We thank the reviewer for detailed comments. In this work, we updated the emission and the ageing scheme in RegCM and examined the model improvement. Our simulations also examined the changes in model outputs just because of change in emission and just because of change in ageing scheme. We did not include these intermediate results in our earlier version and our main message was that climate models require both emission and aerosol processes to be updated in order to improve their performances. However, as pointed out by this reviewer and other reviewers, we agree that it is important to show these intermediate results (now shown in supplementary figures S2, S3, S16, and S17). We want to add that this is not just sensitivity study, rather it implements a new ageing scheme in RegCM, and then use the modified model to simulate with better emission estimates to understand how the model performance behaves.

[Figure]

Figure S2 Spatial patterns of mean seasonal surface BC concentration (µg m$^{-3}$) over India (1st column) using the default set-up and percentage differences in the (2nd and 3rd columns) modified and (4th column customized configurations relative to the default set-up.

[Figure]

Figure S3 Spatial patterns of mean seasonal surface OC concentration (µg m⁻³) over India (1st column) using the default set-up and percentage differences in the (2nd and 3rd columns) modified and (4th column customized configurations relative to the default set-up.

[Figure]

Figure S16. Seasonal variation of vertically distributed mass concentration (µg m⁻³) of BC over the highly polluted Indo-Gangetic Plain

[Figure]

Figure S17. Seasonal variation of vertically distributed mass concentration (µg m⁻³) of OC over the highly polluted Indo-Gangetic Plain

1. **"The ageing changes should change the ratio of dry deposition/wet deposition in the model and later the ageing the lifetime of aerosols in the model. There are no figures in the paper that show the deposition fluxes and their changes and I am left to wonder if that was even significant in the results? As the primary idea seems to be that this will change the lifetime of BC and OC in the model, why are there no calculations of lifetime of particles regionally, seasonally or annually?"**

*Response:* The ageing does change the dry and wet deposition of the tracers and due to this change the ratio of hydrophobic to hydrophilic changes which in turn is altering the atmospheric lifetime of aerosols. The detailed explanation regarding these changes due to ageing alone can be referred from Ghosh et al. 2021. Seasonal variation of lifetime of particles, at the surface and upper atmosphere, due to ageing alone have been already explained in Ghosh et al. 2021.

[Figure]

Seasonal variation of ageing time-scale anomaly (in h) of carbonaceous aerosols at 1000 hPa w.r.t the fixed ageing time-scale of 27.6 h [1.15 day]. The upper level of the colour scale bar has been capped at 140 h (i.e. the ageing time scale is (27.6+140) h =167.6 h or 7 days) and the lower limit has been capped at -20 h (i.e. the ageing time scale is (27.6 – 20) h = 7.6 h). The figure shows the complete domain of simulation, and the area within the black rectangle represents the study domain. (Ghosh et al. 2021, JGR)

**2. "Are there any changes because of the lifetime changes in particle fluxes over to the Indian Ocean that varies by season?"**

_**Response:**_ Since the focus of this manuscript is Indian landmass only, the changes in aerosol properties over the oceans have not been discussed. The oceanic condition is mostly clean with a low concentration of tracers, compared to that over the landmass. There are hardly any emissions over the oceans, as shown in the emission plots (supplementary material Fig S1). In Ghosh et al. 2021 it is evident that the ageing time of the carbonaceous aerosols over the oceans is larger than the default ageing timescale. The explanation has been updated in the revised manuscript in section 2.1 (lines 155-160).

[Figure]

Seasonal variation of ageing time-scale anomaly (in h) of carbonaceous aerosols at 1000 hPa w.r.t the fixed ageing time-scale of 27.6 h [1.15 day]. The upper level of the colour scale bar has been capped at 140 h (i.e. the ageing time scale is (27.6+140) h =167.6 h or 7 days) and the lower limit has been capped at -20 h (i.e. the ageing time scale is (27.6 – 20) h = 7.6 h). The figure shows the complete domain of simulation, and the area within the black rectangle represents the study domain. (Ghosh et al., 2021).

3. **"There are only 17 levels in the models, not sure how many of them are in the boundary layer?"**

*Response:* There are 3 levels (1000, 925, 850 hPa) within the boundary layer and has been added in the revised manuscript (Section 2.1, lines – 143-144).

4. **"How well does the model simulate convection and mixing over the Ganges valley and central plains where the transport to above PBL could be a major factor in increasing the long-range transport of aerosols and hence their lifetimes?"**

**Response**: The model simulated convective tendency and lateral advection (responsible for long-range transport) are given below. More positive values indicate strong updraft above the surface due to convection. Convection tendency gradually increases from left to right (Figure 1.1). Particularly, in the drier seasons since more particles are available in absence of washout. During winter, augmented model (Dyn_regio) is showing lesser pumping effect over IGP than that when only emissions have been changed (Fix_regio). This can be due to the fact that, in presence of dynamic ageing a greater number of hydrophilic tracers are available for removal (evident from the removal plot of BC_HL) even for small amount of precipitation from western disturbances. However, during post-monsoon (OND), due to negligible precipitation over IGP, removal rates of hydrophilic tracers are comparable and hence the pumping effect also follows the same trend. Similar trend in convective tendency is also shown by OC particles (Figure 1.2). The magnitude of OC convection tendency is stronger than that of BC particles. This can be due to the higher concentration of available particles.

Lateral advection on the other hand is an indicator of horizontal long-range aerosol transport. More positive values indicate strong flow along the surface due to advection. Advection shows strong seasonality (from top to bottom – Figure 1.3). In drier months (JF and OND) horizontal transport is comparatively less than pre-monsoon (MAM) and monsoon (JJAS). Therefore, vertical convection is more prominent in dry seasons while horizontal advection is dominant for MAM and JJAS, irrespective of the choice of schemes. Consequently, the observed BC concentration is due to convection in JF and OND and due to advection in MAM and JJAS. Same logic can be applied for OC concentration distribution due to lateral advection (Figure 1.4). However, the positive advection signal is stronger than that of BC particles. This can be again due to the higher concentration of available particles for transport to other regions. Figures for BC convective tendency and lateral advection have been added in the main revised manuscript (Figure 5 and Figure 6) and that for OC has been in the revised supplementary document (Figure S11 and S12).

[Figure]

Figure: 1.1: Seasonal distribution of convective tendency (kg/kg/sec) of BC over IGP for four distinct experiments.

[Figure]

Figure: 1.2: Seasonal distribution of convective tendency (kg/kg/sec) of OC over IGP for four distinct experiments.

[Figure]

Figure: 1.3: Seasonal distribution of lateral advection (kg/kg/sec) of BC over IGP for four distinct experiments.

[Figure]

Figure: 1.3: Seasonal distribution of lateral advection (kg/kg/sec) of OC over IGP for four distinct experiments.

**5. "How well does the model simulate column water depths and hence removal rates through wet deposition in the model?"**

Response: The column water depths are not stored as a model output and hence not included in the response document. The wet removal process of the model has been already explained in the submitted manuscript in Section 2.1 (line 120-129):

"Wet deposition in the RegCM4 has been split into "in-cloud" and "below-cloud" terms. The in-cloud removal process starts for large scale clouds if the liquid water is higher than the threshold level (0.01 g m$^{-3}$) in the model layers where the cloud fraction is more than zero and is a function of the fractional removal rate of liquid water (fraction of precipitating rain over liquid water content of the atmospheric layer, the in-cloud removal rate for cumulus clouds is constant and fixed at 0.001 s$^{-1}$) and the aerosol solubility. This solubility is different for different species, and thus hydrophilic, and hydrophobic BC/OC have different in-cloud wet deposition rates. The below-cloud washing out of the aerosols is controlled by their effective diameters and densities. Collection efficiency for each aerosol species is computed from the aerosol effective diameter and density, which is different for different species."

Since observation measurements of wet removal are not available, direct comparison to quantify the removal is not possible. The model simulated seasonal distribution of BC_HB and BC_HL removal rates are given below for distinct four experiments and can be added in the revised supplementary document.

It can be observed that for BC_HB, maximum removal can be observed by wet removal during JJAS for all the four experiments with varying magnitudes (highest being in Expt Dyn_reg). During drier months, BC_HB dry removal is observed to be most prominent in Expt Fix_reg and Dyn_reg and clearly due to more emissions. Because in Expt Dyn_global dry removal is lowered due to more conversion to HL (Ghosh et al., 2021). Next in case of BC_HL, removal is highest during JJAS.

However, during drier months, both wet deposition and dry deposition are showing signals due to higher availability. This availability results either due to faster conversion or higher emission or the combined effect. For dry removal process, it is due to the negligible precipitation. But. for wet removal it is due to the fact that even for small amount of precipitation more particles are available for removal. Similar results are available for OC as well but not shown here. Wet removal results for BC_HL and OC_HL are shown in the supplementary information (Figure S7 and S8).

[Figure]

Figure 1.4: Seasonal distribution of BC_HB dry removal for four distinct experiments

[Figure]

Figure 1.5: Seasonal distribution of BC_HB wet removal for four distinct experiments

[Figure]

Figure 1.6: Seasonal distribution of BC_HL dry removal for four distinct experiments

[Figure]

Figure 1.7: Seasonal distribution of BC_HL wet removal for four distinct experiments

6. **"The radiative forcing calculations are for direct radiative forcing one assumes as there is no discussion of aerosol-cloud interactions in the model. If that is the case, did you separate the clear days from cloudy days to perform these forcing calculations are these are seasonal averages for days with and without clouds?"**

**Response**: Currently the model does not assume aerosol interaction with clouds, therefore the radiative feedback is mainly governed by direct radiative forcing. Cloud parameters in the model change in response to the radiative feedback. Therefore, we didn't separate clear and cloudy days. This limitation is explicitly mentioned in the revised manuscript in section 3.3 (lines 478-479).

7. **"How well does the model represent RH in the vertical column?"**

**Response**: The model is able to capture the both seasonality as well as vertical profile of relative humidity (RH) in Figure 1.9 and Figure 1.10 over IGP and PI respectively. Relative humidity from ERA-interim dataset at 1.5degree resolution has been incorporated to compare the model simulated RH values (Figure 1.8). During monsoon, the RH is highest (>70%) throughout the troposphere up to 500hPa over both IGP and PI. This is followed by post-monsoon (OND), winter (JF) and pre-monsoon (MAM). During, post monsoon the near surface high RH values over IGP facilitate the fog formation (Dey, 2018; Chakraborty et al., 2016). With gradual decrease in atmospheric moisture, RH values decreases in winter and is mainly dominated by the one brought in by western disturbance over IGP. Finally, the lowest values are captured during MAM. Over PI, high RH values have been captured throughout the year with maximum during monsoon (JJAS) followed by post-monsoon. The RH values during monsoon is due to the south-west monsoon and for post-monsoon it is the north-east monsoon. The fig1.9 and fig1.10 have been incorporated in the revised supplementary document fig S14 and fig S18 respectively.

[Figure]

Figure 1.8: Seasonal distribution of relative humidity (%RH) from ERA-interim dataset over Indo-Gangetic Basin (IGP) and Peninsular India (PI)

[Figure]

Figure 1.9: Seasonal distribution of relative humidity (%RH) over Indo-Gangetic Basin (IGP) for the four distinct experiments.

[Figure]

Figure 1.10: Seasonal distribution of relative humidity (%RH) over Peninsular India (PI) for the four distinct experiments.

8. **"How much effect does the change in aerosol burden in the column have on the atmospheric profiles and how much of this contributes to changes in optical properties and hence forcing?"**

*Response:* The relative humidity (Figure 1.9 for IGP and Figure 1.10 for PI), temperature (Figure 1.11 for IGP and Figure 1.12 for PI) and vertical wind (Figure 1.13 for IGP and Figure 1.14 for PI) profiles have been shown. In terms of change in temperature profile higher temperatures over IGP during MAM and JJAS facilitated the strong vertical wind movement (negative values in Figure 1.13). But negative convective tendency and positive lateral advection of carbonaceous aerosols during these months lowered their concentrations. This is further supported by the high RH values particularly in JJAS which resulted in higher removal. Exactly opposite is happening during the drier months (JF and OND). Comparatively low temperatures, facilitated more stable wind movement (positive values in Figure 1.13). However, in presence of high emissions, the aerosol pumping effect resulted in strong convective tendency which further facilitated the higher concentrations during these months. The low RH values during these are also conducive of higher aerosol atmospheric lifetime. The temperature and vertical wind, omega, profile have been shown in the supplementary information (S13 and S20 for omega in m s$^{-1}$ and S15 and S19 for temperature in °C).

[Figure]

Figure 1.11: Seasonal distribution of temperature (in °C) over IGP for four distinct experiments.

[Figure]

Figure 1.12: Seasonal distribution of temperature (in °C) over PI for four distinct experiments.

[Figure]

Figure 1.13: Seasonal distribution of vertical wind, omega (in m/sec) over IGP for four distinct experiments.

[Figure]

Figure 1.14: Seasonal distribution of vertical wind, omega (in m/sec) over PI for four distinct experiments.

References:

Chakraborty, A., Gupta, T., and Tripathi, S. N.: Combined effects of organic aerosol loading and fog processing on organic aerosols oxidation, composition, and evolution, Science of The Total Environment, 573, 690–698, https://doi.org/10.1016/j.scitotenv.2016.08.156, 2016.

Dey, S.: On the theoretical aspects of improved fog detection and prediction in India, Atmospheric Research, 202, 77–80, https://doi.org/10.1016/j.atmosres.2017.11.018, 2018.

Ghosh, S., Riemer, N., Giuliani, G., Giorgi, F., Ganguly, D., and Dey, S.: Sensitivity of Carbonaceous Aerosol Properties to the Implementation of a Dynamic Ageing Parameterization in the Regional Climate Model RegCM, 126, e2020JD033613, https://doi.org/10.1029/2020JD033613, 2021.

1. **After reading a manuscript, it is unclear whether the authors are arguing that replacing the emission inventory or dynamical ageing scheme or combined effect causes a better representation of carbonaceous aerosols in the RegCM4.6. It will be more appropriate to present or at least discuss (a) how the use of new inventory to represent carbonaceous aerosols improved the simulations? (b) How did implementing the dynamical ageing scheme improve aerosols' representation with default emission inventory and (c) the combined effect? It is recommended that authors design the experiments to address the concerns above.**

*Response:* We thank the reviewer for detailed comments. However, we want to mention that the experiments designed for the manuscript are already as per the suggestion by the reviewer in the section 1 (lines 87-92):

"We carry out four sets of simulations for the year 2010 - (1) control simulation with the default (fixed) ageing scheme and global inventory (hereafter Default_Sc), (2) simulation with the dynamic ageing scheme and global inventory (Dyn_global), (3) simulation with the default ageing scheme and regional inventory (Fix_Regio) and (4) simulation with the dynamic ageing scheme and regional emission inventory (Dyn_Regio)."

We did not include these intermediate results in our earlier version and our main message was that climate models require both emission and aerosol processes to be updated in order to improve their performances. However, as pointed out by this reviewer and other reviewers, we agree that it is important to show these intermediate results (now shown in supplementary figures S2, S3, S16, and S17) and discuss them.

[Figure]

Figure S2 Spatial patterns of mean seasonal surface BC concentration (μg m$^{-3}$) over India (1st column) using the default set-up and percentage differences in the (2nd and 3rd columns) modified and (4th column customized configurations relative to the default set-up.

[Figure]

Figure S3 Spatial patterns of mean seasonal surface OC concentration (µg m⁻³) over India (1st column) using the default set-up and percentage differences in the (2nd and 3rd columns) modified and (4th column customized configurations relative to the default set-up.

[Figure]

Figure S16. Seasonal variation of vertically distributed mass concentration (µg m⁻³) of BC over the highly polluted Indo-Gangetic Plain

[Figure]

Figure S17. Seasonal variation of vertically distributed mass concentration (µg m⁻³) of OC over the highly polluted Indo-Gangetic Plain

2. **From Figures 2 and 3, it is evident that both default and changed models underestimate the surface BC compared to in-situ observations even though the augmented model has better skill than the default one. It is not clear that the better representation of the surface of aerosols (Figure 1) causes more concentration of BC due to convection (vertical mass flux) or lateral advection at higher levels from sources at higher elevations in the IGP region (Fig 4) and why there are no appreciable changes seen over PI region. The in-depth analysis is required to quantify the changes in the vertical of the results in terms of changes in the vertical mass fluxes and vertical velocity vs. mass advection simulated in the model to quantify the impact of on distribution of aerosols.**

_**Response:**_ The augmented model displayed better skill in simulating surface BC over IGP. This is due to the dual role of higher convective tendency and lower lateral advection and vice-versa depending on the season.

Convective tendency is an important indicator for upper air transport of aerosols. More positive values indicate strong updraft above the surface due to convection. Convection tendency gradually increases from left to right (Figure 2.1). Particularly, in the drier seasons since more particles are available in absence of washout. During winter, augmented model (Dyn_regio) is showing lesser pumping effect over IGP than that when only emissions have been changed (Fix_regio). This can be due to the fact that, in presence of dynamic ageing a greater number of hydrophilic tracers are available for removal (evident from the removal plot of BC_HL) even for small amount of precipitation from western disturbances. However, during post-monsoon (OND), due to negligible precipitation over IGP, removal rates of hydrophilic tracers are comparable and hence the pumping effect also follows the same trend. Similar trend in convection tendency is also shown by OC particles (Figure 2.2). The magnitude of OC convection tendency is stronger than that of BC

particles. This can be due to the higher concentration of available particles. Figures for BC convective tendency and lateral advection have been added in the main revised manuscript (Figure 5 and Figure 6) and that for OC can be added in the revised supplementary information (Figure S11 and S12).

[Figure]

Figure 2.1: Seasonal distribution of convective tendency (kg kg⁻¹ sec⁻¹) of BC over IGP for four distinct experiments.

[Figure]

Figure 2.2: Seasonal distribution of convective tendency (kg kg⁻¹ sec⁻¹) of OC over IGP for four distinct experiments.

Lateral advection on the other hand is an indicator of horizontal long-range aerosol transport. More positive values indicate strong flow along the surface due to advection. Advection shows strong seasonality (from top to bottom – Figure 2.3). In drier months (JF and OND) horizontal transport is comparatively less than pre-monsoon (MAM) and monsoon (JJAS). Therefore, vertical convection is more prominent in dry seasons while horizontal advection is dominant for MAM and JJAS, irrespective of the choice of schemes. Consequently, the observed BC concentration is due to convection in JF and OND and due to advection in MAM and JJAS. Same logic can be applied for OC concentration distribution due to lateral advection (Figure 2.4). However, the positive advection signal is stronger than that of BC particles. This can be again due to the higher concentration of available particles for transport to other regions.

[Figure]

Figure 2.3: Seasonal distribution of lateral advection (kg kg$^{-1}$ sec$^{-1}$) of BC over IGP for four distinct experiments.

[Figure]

Figure 2.4: Seasonal distribution of lateral advection (kg kg⁻¹ sec⁻¹) of OC over IGP for four distinct experiments.

Over PI, the lower concentration can be primarily because of the lower emissions for both BC and OC. Convective tendency as well as lateral advection for BC is not playing any major role (as can be seen in Figure 2.5 and 2.7), hence concluding the role of lower emissions. In case of OC, lateral advection (Figure 2.6 and 2.8) and comparatively lower emissions (Figure in the actual supplementary document) than IGP can be the predominant factors for lower concentration over PI in presence of negative convective tendency.

[Figure]

Figure 2.5: Seasonal distribution of convective tendency (kg kg$^{-1}$ sec$^{-1}$) of BC over PI for four distinct experiments.

[Figure]

Figure 2.6: Seasonal distribution of convective tendency (kg kg$^{-1}$ sec$^{-1}$) of OC over PI for four distinct experiments.

[Figure]

Figure 2.7: Seasonal distribution of lateral advection (kg kg$^{-1}$ sec$^{-1}$) of BC over PI for four distinct experiments.

[Figure]

Figure 2.8: Seasonal distribution of lateral advection (kg kg⁻¹ sec⁻¹) of OC over PI for four distinct experiments.

3. **The result in Lines 371-3, "Due to the model improvements (forcing estimates 371 with the default model are shown in Figure S8), the TOA forcing changes by -72.75%, and the 372 surface dimming increases by 39.73% over the IGP and by -23.94% and 34.35%, respectively," should be cross-checked with the amount of clouding simulated model and reflections from clouds at TOA due to them vs. the effect of surface dimming as mentioned in the manuscript to be sure. Alternatively, these differences in the simulations can be attributed to the amount of cloudiness simulated (secondary effects) by default and the augmented model.**

_**Response:**_ Currently the model does not assume aerosol interaction with clouds, therefore the radiative feedback is mainly governed by direct radiative forcing. Therefore, secondary effects due to aerosols cannot be considered for the observed values. Since the model does not consider aerosol-cloud interactions, it is explicitly mentioned in the manuscript in section 3.3 (lines 460-462). However, the amount of cloudiness simulated by the model in four distinct experiments is given below. It can be seen that the model is able to capture the seasonal variability of total cloud cover for each of the four experiments. The amount of cloud cover is maximum during monsoon. The cloud cover is low during other months. Thus, the reflection from clouds will also be lower. As a result, contribution to the observed anthropogenic AOD due to cloud reflections will also be lower. Therefore, AAOD distribution over IGP is primarily responsible for the surface dimming effect and the resulting atmospheric heating.

[Figure]

Figure 2.9: Seasonal distribution of total cloud fraction (in %) for four distinct experiments.

**4. The manuscript will be more readable if the same terminology is used in the revision to specify model setup (augmented model or customized setup).**

*Response:* The terminology has been updated to "augmented model" throughout the manuscript as per the suggestion.

**5. Line 402-403 "Our work demonstrates that even the improvement of some aspects of the aerosol representation can lead to substantial enhancements in the model performance." The sentence requires to be rewritten with more quantification and elaboration.**

Response: In the supplementary figure S4, the quantification of model performance has been already shown. At most of the in-situ sites out of 24, only dynamic ageing implementation resulted in 5-10 % improvement. But when both regional emissions along-with ageing is implemented, the model representation of BC surface concentration (µg/m3) increased by 60-120%, particularly for polluted sites like Patiala, Kanpur, Varanasi.

[Figure]

Figure S4. Locations of the 24 cities where BC concentrations were measured during the study period and used to evaluate the customized model performance. The colour of the circles indicates the percentage increase in BC concentrations due to the implementation of the dynamic scheme and the size of the circles indicate the percentage increase in BC concentrations due to the combined impact of ageing scheme and regional inventory in the customized model.

6. **More justification is needed to conclude that " a dynamic ageing scheme and a regional emission inventory substantially improve the model performance over the Indian sub-continent." and "The BC and OC surface concentration and column burden increase due to the model improvements, more so as a combined effect of the two factors than because of the individual ones."**

Response: The following figure further justifies the conclusion: " a dynamic ageing scheme and a regional emission inventory substantially improve the model performance over the Indian sub-continent." In each of the season particularly in winter (63.54%), the mean BC burden in 2.10(a) is showing maximum improvement for Dyn_reg experiment w.r.t to default. Similar, increments are visible for BC surface concentration as well (winter is showing maximum change of 61.46%).

[Figure]

[Figure]

Figure 2.10: Seasonal distribution of % change of (a) BC burden (mg/m2) and (b) BC surface concentration (µg/m3) for each sensitivity experiment w.r.t the default set-up where De = Default, DG = Dyn_global, FR = Fix_reg and DR = Dyn_reg.

---

## Author Response (AR2)

**Responses to Editorial Comments:**

The authors have successfully addressed the concerns raised by the two reviewers. The revised manuscript is scientifically sound and contributes to the carbonaceous aerosol modeling field over the Indian monsoon region. Therefore, I recommend its publication after considering the following suggestions for technical corrections.

*Response: We thank the Editor for recommending publication after the technical corrections. We have modified the manuscript based on the suggestions. The revised sections are highlighted.*

1/ The model evaluation results of the different sensitivity simulations clearly shown that the dominant source of model performance improvements is the use of the regional emission inventory, while the dynamic ageing scheme only has a marginal effect. This should be emphasized throughout the text, including the abstract and the conclusions.

*Response: This has been emphasized throughout, including Abstract and Conclusions.*

2/ The figure used on the response to the 6th comment of the 2$^{nd}$ reviewer illustrates excellently how the two developments/modifications contribute to the overall improvement of the model performance. I suggest to include this figure on the main text.

*Response: This figure is now included in the min manuscript (Figure 9).*

3/ Lines 37-39 and lines 519-521: I suggest to rephrase these two statements since they are misleading. While both modifications lead to improvements on the model performance, clearly the emissions are far more important.

*Response: We rephrased these statements in the revised version.*